# The Role of Sex in the Impact of Sleep Restriction on Appetite- and Weight-Regulating Hormones in Healthy Adults: A Systematic Review of Human Studies

**DOI:** 10.3390/clockssleep7030039

**Published:** 2025-07-29

**Authors:** Mira Alfikany, Khaula Sakhr, Stef Kremers, Sami El Khatib, Tanja Adam, Ree Meertens

**Affiliations:** 1Department of Health Promotion, NUTRIM Institute of Nutrition and Translational Research in Metabolism, Maastricht University Medical Center+, P.O. Box 616, 6200 MD Maastricht, The Netherlands; m.alfikany@maastrichtuniversity.nl (M.A.); s.kremers@maastrichtuniversity.nl (S.K.); 2Department of Nutrition and Food Sciences, School of Arts and Sciences, Lebanese International University, Bekaa, Beirut P.O. Box 146404, Lebanon; khaula.s8@hotmail.com; 3Department of Biomedical Sciences, School of Arts and Sciences, Lebanese International University, Bekaa, Beirut P.O. Box 146404, Lebanon; sami.khatib@liu.edu.lb; 4Center for Applied Mathematics and Bioinformatics (CAMB), Gulf University for Science and Technology (GUST), Mubarak Al-Abdullah 32093, Kuwait; 5Department of Nutrition and Movement Sciences, NUTRIM Institute of Nutrition and Translational Research in Metabolism, Maastricht University, P.O. Box 616, 6200 MD Maastricht, The Netherlands; t.adam@maastrichtuniversity.nl; 6Department of Health Promotion, NUTRIM Institute of Nutrition and Translational Research in Metabolism, and Care and Public Health Research Institute (CAPHRI), Maastricht University, P.O. Box 616, 6200 MD Maastricht, The Netherlands

**Keywords:** sleep, restriction, deprivation, sex differences, hormones, appetite

## Abstract

Short sleep has been linked to overweight, possibly via alterations in appetite-regulating hormones, but findings are inconsistent. Sex differences may contribute to this variability. This systematic review examines whether sex modifies the hormonal response to sleep curtailment. PubMed, Embase, Cochrane, CINAHL, and PsycINFO were searched for English-language experimental studies published before December 2024. Included studies assessed at least one appetite-regulating hormone and presented sex-specific analyses. Studies involving health conditions affecting sleep, circadian misalignment, or additional interventions were excluded. Risk of bias was assessed using the Revised Cochrane Risk-of-Bias tool (RoB 2). Eight studies (*n* = 302 participants) met inclusion criteria. A narrative synthesis of the findings was conducted for each hormone separately to explore potential differences in their response to sleep restriction. Some sex-related variations in hormonal response to sleep restriction have been observed for leptin (four studies, *n* = 232), insulin (three studies, *n* = 56), glucagon-like peptide-1 (one study, *n* = 27), ghrelin (three studies, *n* = 87), adiponectin (two studies, *n* = 71) and thyroxine (two studies, *n* = 41). However, findings were inconsistent with no clear patterns. No sex-related differences were found for glucagon or PYY, though data were limited. Findings suggest sex may influence hormonal responses to sleep restriction, but inconsistencies highlight the need to consider factors such as BMI and energy balance. Well-controlled, adequately powered studies are needed to clarify these effects.

## 1. Introduction

Sleep is vital for life, a cornerstone of both physical and mental health. It improves mood, cognitive skills and metabolic health [1]. However, the global prevalence of sleep disturbances and sleep--related disorders has increased significantly over the last decades [2,3]. The National Sleep Foundation (NSF), the American Academy of Sleep Medicine and the Sleep Research Society recommend adults sleep 7 h or more per day [4,5]. Sleep deprivation is defined as the inability to achieve the necessary amount of sleep [6] and is due to partial sleep restriction (SR) or total sleep deprivation (TSD) [7]. Sleep debt is associated with many adverse health outcomes, impaired well-being and cognitive performance, and decreased work productivity [8,9,10]. A population particularly at risk includes school and university students who struggle with early schedules, especially as individuals of this age have later chronotypes [11]. Those individuals often report poor sleep quality, greater fatigue and less regular social rhythms [12]. Voluntary and/or experimental sleep restriction has been previously linked to adverse metabolic and cardiovascular health effects [13,14,15] and increased body weight [11,12]. Results concerning health outcomes, however, are rather inconsistent and are different for males and females [16]. While research indicates that females have consistently reported lower sleep quality in comparison to their male counterparts, objective measurements, including polysomnographic data, described females as exhibiting higher sleep quality and longer total sleep duration compared to males [17,18].

In light of the rising incidence of sleep-related disorders and obesity, the correlation between reduced sleep duration, body weight, and obesity has garnered significant attention over the past few decades. This relationship has been the subject of numerous observational and experimental studies. It has been suggested that the relationship may be partially due to different releases of appetite-regulating hormones, although results are inconsistent [19,20]. These hormones are classified as either orexigenic (stimulating appetite) or anorexigenic (suppressing appetite). The most well-known orexigenic hormone is ghrelin, which is produced by the stomach and stimulates appetite by acting on the hypothalamus [21]. Anorexigenic hormones may act in the short term—stimulated in response to a meal—such as GLP-1 and PYY, both secreted by intestinal L-cells [22], or in the long term, like insulin and leptin, which are influenced by the individual’s adiposity status [22,23]. With regard to other hormones, such as adiponectin, glucagon, and thyroid hormones (T3 and T4), the literature reports conflicting results, depending on certain conditions, concerning their role in regulating energy metabolism and appetite [24,25]. It is widely recognized that sex can affect baseline as well as postprandial concentrations of appetite-regulating hormones [26,27]. Taking into account an equivalent quantity of adipose tissue, females were shown to have higher leptin concentrations and lower insulin concentrations compared to males [28]. Although the role of adiponectin for food intake regulation may be debatable, baseline concentrations of adiponectin tended to be higher in females compared to males [29]. Moreover, following the consumption of a well-balanced mixed meal, ghrelin suppression was shown to be greater in females compared to males [30]. Ghrelin is considered a reverse adiposity signal lacking variability with increased BMI; concentrations were shown to drop in response to weight gain while they increased after weight loss [31]. In individuals with obesity, the suppression of ghrelin following meals is significantly compromised [31]. In a review published in 2023, Chaput suggested that the effect of sleep restriction on the 24 h secretion of appetite hormones, including glucagon-like peptide-1 (GLP-1) and pancreatic polypeptide, is potentially dependent on sex in addition to the context of the participants. In another review, Gallegos also suggested sex to be a possible factor in modifying the hormonal response to sleep restriction that warrants further exploration [32].

While previous systematic reviews examined the effect of sleep duration on some weight and appetite-regulating hormones [33,34,35,36,37], the impact of sex specifically on the relationship between sleep duration and appetite hormones has not been addressed in humans. Consequently, this review seeks to address the existing gap by evaluating potential sex-specific associations between sleep curtailment and hormones that are pertinent to the regulation of weight and appetite in healthy adults. Highlighting a possible impact of sex on the relationship between sleep duration and appetite regulation is a crucial step in the development of sex-specific approaches to alleviate the metabolic impact of sleep curtailment.

## 2. Methods

### 2.1. Search Strategy and Eligibility Criteria

This systematic review was conducted using the Preferred Reporting Items for Systematic Reviews and Meta-Analyses (PRISMA) guidelines [38]. The adopted protocol has been registered on Prospero (www.crd.york.ac.uk; ID CRD42020170398, accessed on 1 March 2025). The following databases were explored for articles published from inception until December 2024: PubMed, Embase, Cochrane, CINAHL and PsycINFO. The search prompt, organized in two groups of search words (i.e., sleep and appetite-regulating hormones), included all terms in titles and abstracts and has been conducted independently by two different authors of the study (MA and KS); all discrepancies were discussed and sorted out with RM and TA. The references of included articles were also explored for additional relevant studies. Included hormones comprise leptin, insulin, GLP-1, peptide YY (PYY), ghrelin, glucagon, triiodothyronine (T3), thyroxine (T4) and adiponectin. The search strategies used for all the databases are found in Appendix A.

Eligible studies were experimental studies (randomized controlled trials and parallel design studies) including a sleep duration restriction performed in healthy individuals with at least one measured hormone relevant to appetite regulation. The studies were required to encompass both female and male participants, with the analysis and reporting of appetite hormone concentrations conducted separately for each sex. Only English-language written articles were included. There was no restriction in terms of the duration of the intervention or the number of included participants. Furthermore, all participants, irrespective of their usual sleep duration, were included in the study. Both free-living and laboratory-based research were taken into account. Studies performed on individuals having any physiologic condition or health problem that may affect sleep (pregnancy, diabetes, cancer, cardiovascular diseases, sleep apnea, Parkinson’s, depression, Alzheimer’s, type 1 diabetes, or obstructive sleep apnea) were excluded. In addition, studies including a circadian misalignment or shift work manipulation were not considered. Finally, all studies including an intervention other than sleep restriction where the effect on hormones cannot be separated from the effect of sleep duration were removed.

### 2.2. Data Extraction Process

For the set of studies included in this systematic review, a data collection form was prepared where data extraction and recording were completed by two reviewers (MA and KS) independently. Any discrepancy was resolved by discussion with RM and TA. Extracted data included the following variables: study citation, country of origin, participants’ characteristics (age, BMI/weight), sample size per sex (including the total number of females and males whose data are included for analysis), short description of the sleep restriction, energy balance of participants during the study, duration of washout period in crossover trials, study setting, study design, sex-specific outcome(s) related to the studied hormone(s) (increase or decrease)/significant result(s) (mean, SD, effect size, period effect for crossover trials) and summarized results.

### 2.3. Risk of Bias Assessment

Two authors (MA and KS) independently reviewed the included studies and checked for possible risk of bias. Randomized trials (crossover and parallel design) were assessed using the Revised Cochrane Risk-of-Bias tool for randomized trials (RoB 2) [39]. Discrepancies were resolved by discussion with RM and TA until consensus was reached.

## 3. Results

In total, 30,724 articles were retrieved, of which 14,452 duplicates were eliminated. After identifying the relevant articles and conducting a full-text screening, only eight studies were identified to include relevant data on sex differences and were deemed suitable for data extraction (see the Prisma Flow Chart in Figure 1). Among the eight included interventional studies, seven were randomized controlled trials and one was a parallel design study. The number of participants ranged from 11 to 145. Reported mean ages of participants were variable and ranged from 20.6 ± 1.3 years to 39 ± 5 years, while their BMI ranged from 17.7 to 32.6 kg/m^2^. While seven of the included studies addressed SR, only one study investigated TSD [40]. The duration of sleep restriction varied across studies, ranging from three nights to six weeks, while the TSD intervention was limited to just one night. The level of sleep restriction also varied across studies, with sleep duration ranging from approximately 4 to 5.5 h per night or involving reductions relative to individual baseline sleep. In six of the included studies, the sleep intervention took place in the laboratory. Caloric intake was restricted in some studies, while in others, ad libitum caloric consumption was allowed. Results of the risk of bias assessment are shown in Figure 2. A majority of the studies exhibited certain concerns primarily associated with the randomization process; however, they were generally regarded as high quality.

## 4. Sleep Restriction Effect on Appetite Regulating Hormones

The sex-specific effects of the sleep restriction on appetite-regulating hormones are depicted in Table 1. The table categorizes the studies based on the hormone(s) examined. Results for each sex are reported separately; when data for sex-specific differences are unavailable (e.g., in cases where no sex-specific differences are observed), results for both sexes combined are presented. However, it should be noted that only one study was initially designed and adequately powered to assess sex-specific differences [41]. Therefore, a note has been added alongside the studies to highlight this limitation.

### 4.1. Insulin and Insulin Sensitivity

Three studies addressed sex-related variations in insulin concentrations. Fasting insulin delivered inconsistent results, with one study showing a decrease after SR in females but not in males [42], while another study showed increased concentrations in both sexes with no significant difference between them [43]. Whole-body insulin sensitivity was shown to be significantly decreased after SR by 25% compared to regular sleep, with no significant differences between females and males; males, however, showed a more robust decrease in insulin sensitivity [44]. In summary, the findings pertaining to sex differences in insulin response to SR exhibit a lack of consistency.

### 4.2. Leptin

Four studies reported sex-related variations in leptin concentrations after SR/TSD. St-Onge [42] did not find an effect of SR on fasting leptin concentrations in either females or males. Markwald [45] did not find a significant effect of SR on leptin concentrations in females but did show a significant increase in the average 24 h leptin concentrations in males. Simpson [41] also showed an increase in mean leptin concentrations in both sexes; the effect, however, was more pronounced in females. Finally, Van Egmond [40] found a decrease in fasting leptin in response to TSD in females but not in males. In summary, the findings indicate that there are no consistent results regarding sex differences in the leptin response to sleep restriction.

### 4.3. Ghrelin

Ghrelin was addressed in three of the included studies. After SR, a decrease in average 24 h ghrelin in both females and males was observed by Markwald with no differences for sex [45]. The two other studies observed an increase in ghrelin concentrations after SR and TSD. This increase was seen in males, but not in females, for fasting ghrelin as well as for 24 h total ghrelin concentrations [42]. Van Egmond confirmed the presence of a significant increase in ghrelin concentrations after TSD for both sexes [40]. Based on the findings from these studies, it remains inconclusive to determine the influence of sex on the ghrelin response to stress-related stimuli.

### 4.4. Peptide YY, GLP-1, Adiponectin, T4 and Glucagon

For PYY, two studies addressing this hormone did not show any sex-related differences in its response to SR [42,45]. Regarding GLP-1, one study showed that fasting concentrations were not affected after five nights of SR in neither females nor males. Afternoon concentrations following five nights of SR, however, were shown to be lower in females only [42]. Adiponectin was assessed in two studies; while no significant sex-related differences were found for fasting concentrations after SR, average 24 h concentrations were lower in males only [42]. Another study showed that TSD for one night increased adiponectin concentrations significantly in females only [40]. T4 was assessed in two studies; Kessler showed mean 24 h T4 to be decreased significantly after SR in females, but not in males [46], while Petrov observed that, after SR, T4 increased non-significantly in females and decreased non-significantly in males. Finally, one study assessed the impact of SR on glucagon, showing increased concentrations after three nights of SR in both females and males with no sex differences [43]. In light of the previous literature, it has been shown that sex does not significantly influence the response of both PYY and glucagon to SR. Furthermore, the findings pertaining to GLP-1, adiponectin, and T4 exhibit inconsistencies.

## 5. Discussion

This systematic review aimed at examining the effect of sex as a modifier of the impact of sleep duration restriction on appetite-regulating hormones. The hormones considered were leptin, ghrelin, adiponectin, insulin, glucagon, PYY, GLP-1 and T4. In order to cover all relevant studies, RCTs and parallel design studies that reported results for females and males separately were included in this review. Most studies were randomized controlled trials. A total of 8 studies were included. Most of the sleep deprivation studies were conducted in controlled laboratory settings, while in two studies, participants were allowed to sleep at home [43,47].

Considering the available studies, this systematic review suggests the presence of some sex-related variations in the appetite-regulating hormones response to sleep restriction, although the results are inconsistent. Leptin, insulin, GLP-1, ghrelin, adiponectin and T4 have all shown variations in their response to SR, with no clear or specific patterns emerging. For glucagon and PYY, no sex-related hormonal differences were observed, likely due to the limited number of available studies. The findings indicate that sex alone is insufficient to account for the observed variations in hormonal responses to sleep restriction. It is imperative to consider additional variables that may contribute to these differences, and such factors should be incorporated into future research endeavors. Furthermore, these findings should be cautiously interpreted, as most of the included studies were not originally designed or powered to test for sex-specific variations. Therefore, the presence or absence of such differences may be influenced by limited statistical power rather than by true underlying effects.

Insulin and leptin are adiposity signals regulating food intake and energy expenditure [48]. Insulin resistance, the inability of a given amount of insulin to sufficiently stimulate glucose uptake and utilization, is a hallmark feature of type 2 diabetes [49]. Insulin resistance is more prominent in males compared to females [50]. The current review did not identify any significant differences in fasting concentrations or the overall response to sleep deprivation between female and male subjects. However, it was observed in one study that males exhibited a specific reduction in insulin sensitivity in response to sleep restriction. The sex dimorphism with regard to insulin sensitivity may be associated with sex-specific body fat distribution, differences in adipose lipid metabolism and age differences in the investigated research participants [50].

With regard to ghrelin, sex differences in its relationship with SD were observed. Those differences could in part be related to differences in sex hormones. The synthesis, secretion and degradation of ghrelin, one of the most potent orexigenic peptide hormones, appear to be directly modulated by estradiol, suggesting different impacts on food intake throughout the menstrual cycle [51]. Conversely, estrogen has been shown to directly oppose orexigenic and other metabolic effects of ghrelin in the body [51]. Sex-specific hormones, therefore, may act to conserve ghrelin metabolism in the face of SR for females under certain circumstances [51,52].

GLP-1 is produced by intestinal L-cells in the distal small intestine and in the colon; its secretion is enhanced by estrogen, which also increases GLP-1R sensitivity [53]. Given the influence of estrogen on GLP-1 secretion, changes in GLP-1 response to glucose are expected to be seen between the follicular and luteal phases and between pre- and postmenopausal women [53]. GLP-1 secretion from the colon appears to decline with age [54]. Consequently, considering a diverse age range may reduce the likelihood of identifying differences in the effects of SR. The overall results on GLP-1 and ghrelin suggest that SR may increase appetite for males through increased ghrelin concentrations while it increases appetite for females through a decrease in GLP-1 concentrations, as has been suggested previously [42].

Adiponectin is an adipokine mainly known for its anti-inflammatory, anti-oxidant and insulin-sensitizing roles [55]. It is mainly produced by adipose tissue with concentrations positively associated with gluteofemoral body fat [56]. Sex differences in adiponectin concentrations may in part be explained by changes starting during puberty, where the increase in androgen concentrations in males reduces adiponectin production [57]. The more pronounced reduction in adiponectin concentrations observed in males after sleep restriction may therefore be due to androgen-specific effects. In contrast, in females, estrogen might confer a protective role. It is worth noting that some studies show sleep restriction to be associated with menstrual irregularities and with altered concentrations of estrogen in the body, adding more complexity to the analysis of sex differences in conditions of SR [58].

Another possible explanation for the sex differences concerning the relationship between SR and appetite regulation may be activity of the hypothalamic–pituitary–adrenal (HPA) axis, ultimately serving the purpose of releasing cortisol [59]. Cortisol previously has been linked to elevated ghrelin plasma concentrations and reduced leptin plasma concentrations [60,61]. Given that sleep deprivation serves as a significant stressor leading to elevated cortisol levels, the HPA axis may play a crucial role as a mediator in the sex-dimorphic relationship between sleep deprivation and appetite regulation. Research has shown that reduced subjective sleep quality, diminished sleep efficiency and/or increased awakenings during the night might be related to higher reactivity of the HPA axis [62]. Variations in sleep duration do not seem to enhance the stress responsiveness of the HPA axis [62]. Females appear to be more susceptible to sleep-related changes in the sensitivity of the HPA axis [62,63]. Some support for the role of the HPA axis is coming from a randomized crossover study in men, which demonstrated that increased insulin resistance and hyperinsulinemia due to SR can be mitigated by clamping cortisol and testosterone [64].

While this systematic review highlights important findings supporting the sex differences in the relationship between sleep curtailment and appetite-regulating hormones, inconsistencies in those findings were even more apparent. The variations observed in the protocols of the studies (including the duration and the severity of sleep restriction), the age distribution of the participants, and the specific forms of the hormone assessed may contribute, at least partially, to the inconsistencies found in the results. Age, weight status of participants and detailed information with regard to energy intake and expenditure are factors of particular importance.

The variation in age of included participants, ranging from 18 to over 45 years, reflects the inclusion of young adults as well as middle-aged individuals. This variability may contribute to the heterogeneity observed in the hormonal responses, as age is known to influence metabolic and endocrine functions. Ghrelin and its physiological role change with age; older lean females tend to have lower levels of ghrelin compared to younger lean females and these levels remain relatively constant even in the presence of obesity, unlike in younger females, where ghrelin levels decrease in the presence of obesity [65]. Moreover, significant correlations between ghrelin and other hormones such as leptin and adiponectin, as well as with components of metabolic syndrome, have been observed in young adults but not in older individuals [65]. Furthermore, depending on age, obesity is associated with the development of leptin resistance [66]. In fact, middle-aged individuals usually experience weight gain that is followed by a state of sarcopenia at more advanced ages [66]. However, plasma leptin seems not to be affected by age as much as it is affected by the BMI of individuals. Many studies have shown that plasma leptin levels do either not change or slightly decrease with age, but they tend to increase depending on BMI independent of age [66]. Other published studies have also highlighted associations between age and the levels of serum adiponectin [67], thyroid hormones [68], and GLP-1 and PYY [54]. However, findings have been mixed, indicating the need to consider other factors beyond age in the interpretation of the levels of these hormones.

While most of the studies enrolled individuals with normal weight and overweight, none included exclusively participants of normal weight (BMI < 25 kg/m^2^). Other studies recruited individuals spanning a broader BMI range, from underweight (BMI = 17.7 kg/m^2^) to obese (BMI = 32.6 kg/m^2^) [40,41]. Weight is known to play an important role in ghrelin excursions [69]. Ghrelin concentrations were significantly lower in participants with elevated fat mass compared to healthy weight participants both at baseline and after total sleep deprivation [70]. A recent study conducted by Salem [71] showed adiponectin to be significantly lower in females who are overweight/obese compared to participants of normal weight. Moreover, GLP-1 release in the ileum declined with increasing BMI [54]. Several sleep restriction studies have previously supported the impact of BMI on the hormonal response to sleep restriction [40,70].

Furthermore, the energy balance of participants, along with their dietary intake patterns and diet composition, was not consistently controlled across the studies. Energy intake of participants was controlled in some studies, but not in others, and sometimes not even assessed, which makes it difficult to reach conclusions [41,42,43,44]. Sleep deprivation is characterized by a rapid increase in energy expenditure of about 4–5% that is maintained across many days [72]. This increase in energy expenditure, if not counterbalanced with similar increases in energy intake, will result in a negative energy balance [72]. Broussard [73] suggested that ad libitum or uncontrolled feeding conditions may decrease ghrelin concentrations and consequently mask the effect of sleep restriction. On the other hand, hypocaloric diets that induce moderate weight loss do not necessarily increase ghrelin concentrations [72]. Concerning the studies incorporated within this systematic review, ad libitum food intake led to an increase in serum leptin both in females and males, with a higher effect observed in females in one of the studies [41], while in another one, the increase in leptin following SR seems to have been blunted in females because of the feeding restriction that they experienced [45]. On the other hand, a slight negative energy balance may have eliminated any significant effect of SR on leptin both in females and males [42]. It is also important to note that, following sleep restriction and in cases of uncontrolled feeding, food choices and consumption patterns might change, and individuals tend to choose high-caloric foods high in sugar and fat as a stress-coping strategy [36,74,75]. Therefore, independently from its effect on appetite hormones, sleep restriction contributes to weight gain by increasing the desire to consume unhealthy food choices; in fact, SR activates some brain regions related to hunger, food choices and reward mechanisms, leading people to select high-energy dense foods [72]. Hence, assessing and possibly controlling energy intake in sleep restriction studies might give more homogenous insights into the sex-specific effects on appetite-related hormones.

Although research supports the involvement of different factors in determining the levels of appetite-regulating hormones, our systematic review did not reveal a consistent pattern regarding how these factors influence the effect of sleep restriction on hormonal response. The limited number of available studies, most of which are not adequately powered to test for sex-related differences, along with the heterogeneity in experimental conditions, such as the severity of sleep restriction, the differences in hormone analyses and sampling methods, and the inclusion of participants with wide ranges of age and BMI, make it difficult to draw clear conclusions.

### Strengths and Limitations

To the best of our knowledge, this is the first systematic review to address sex as a possible modifier of the impact of sleep restriction on the concentration of appetite-regulating hormones and to include a multitude of appetite-regulating hormones. By addressing this factor, variables have been highlighted that commonly affect the outcomes in studies of sleep restriction and appetite hormones and that should be considered in future studies addressing this topic. The study also demonstrated several strengths, notably the extensive search conducted across a multitude of databases and the rigorous methodology employed, wherein all phases of the screening and data extraction processes were carried out by two independent researchers.

Nevertheless, it is essential to recognize several limitations that must be acknowledged. A first limitation of this review is the inclusion of only a small number of articles due to strict inclusion criteria as well as the restriction to English-language articles only. The studies included in this review also have limitations; the studies usually have small sample sizes and were not powered to address sex-specific research questions. The duration of the studies was often short, inconsistent methods for measuring sleep and hormones were applied, and most were lab studies, which may limit the generalizability of the findings. Furthermore, there was substantial heterogeneity in sleep restriction protocols, and the limited number of studies using comparable sleep cut-offs precluded meaningful subgroup analyses.

## 6. Conclusions

The present review supports the notion of sex-based differences in the response of several appetite-regulating hormones to sleep curtailment, but findings remain inconsistent. The number of studies addressing sex differences in the literature is limited, and therefore well-controlled, sufficiently powered, and specifically designed studies are needed to come to a substantiated conclusion. To utilize the sex-dimorphic and complex mechanisms underlying the relationship of sleep duration and appetite regulation for sex-specific prevention and treatment, it is of great importance to include both sexes in further studies. Moreover, other relevant variables, including age, BMI and energy balance control of participants, should be considered in those studies.

## Figures and Tables

**Figure 1 clockssleep-07-00039-f001:**
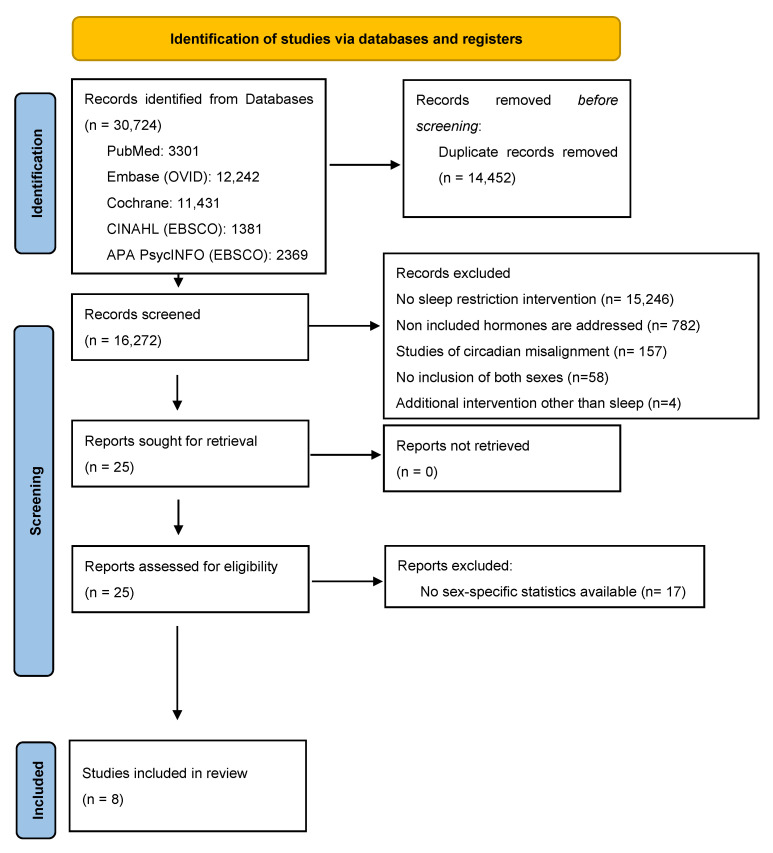
Prisma flow diagram of literature search and articles selection process.

**Figure 2 clockssleep-07-00039-f002:**
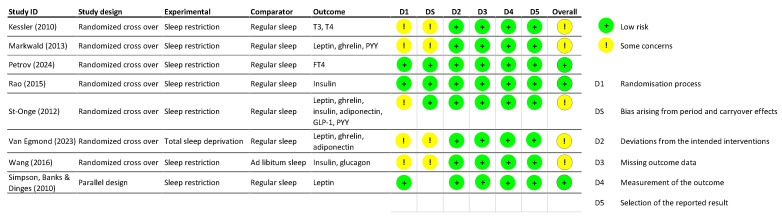
Results obtained for the risk of bias assessment of randomized cross over trials and parallel design studies [40,41,42,43,44,45,46,47].

**Table 1 clockssleep-07-00039-t001:** Results of individual studies presenting sex-specific results for the effect of sleep restriction on appetite- and weight-regulating hormones.

Author (Year)	Country	Participants’ Characteristics [Age (Mean ± SEM), BMI/Weight (Mean ± SEM)]	Sample Size per Sex (Whose Data Are Included for Analysis)	Short Description of the Sleep Restriction	Energy Balance of Participants	Presence/Absence and Duration of Washout Period	Study Setting	Study Design	Sex-Specific Outcome(s) Related to the Studied Hormone(s) (Increase or Decrease)/Significant Result(s) (Mean, SD, Effect Size, Period Effect for Crossover Trials)	Summarized Results (Sex Differences in the Effect of the SR/TSD on the Specified Hormone: None/+/−
**Leptin**										
Markwald, R (2013) [45]	USA	Age: 22.4 ± 4.8 years, BMI: 22.9 ± 2.4 kg/m^2^	16 (males, *n* = 8; females, *n* = 8)	Three baseline days of 9 h/night sleep, then half the participants underwent 5 days of SR (5 h/night), and the other half remained on a 9 h per night sleep. Then participants crossed over to the other condition	Weight maintenance diets during the baseline period. Ad libitum intake during sleep intervention.	ND	In lab	Randomized crossover trial	SR vs normal sleep: Average 24 h leptin levels for both sexes: 6.7 ± 5.1 ng/mL vs. 5.5 ± 5.2 ng/mL, respectively; leptin levels for females were not statistically different across conditions, whereas leptin levels for males were higher in the 5 h condition versus baseline. Not designed to study sex-specific differences.	SR effect on leptin in males: + SR effect on leptin in females: None
St-Onge, MP (2012) [42]	USA	Age: 30–45 years, BMI: 22–26 kg/m^2^	27 (males, *n* = 14; females, *n* = 13)	5 nights of SR (4 h/night) versus 5 nights of habitual sleep (9 h/night)	Controlled diet (30% fat, 55% carbohydrates, and 15% protein) with fixed meal times. Energy requirements were estimated using the Harris-Benedict equation.	3 weeks	In lab	Randomized crossover trial	SR vs normal sleep: Fasting leptin for both sexes: no significant differences Designed to study sex-specific differences.	SR effect on fasting leptin in males: None SR effect on fasting leptin in females: None
Van Egmond, L (2023) [40]	Sweden	Age: 24.9 ± 2.9 years, BMI: 27.8 ± 6.7 (males) vs. 27.0 ± 6.1 kg/m^2^ (females)	44 (males, *n* = 24; females, *n* = 20)	Each participant had 1 night of TSD and 1 night of normal sleep	Standardized dinner followed by an overnight fast	1 week	In lab	Randomized crossover trial	TSD vs normal sleep: Fasting leptin for females: 25.8 ± 4.3 vs. 28.1 ± 4.7 ng/mL, *p* = 0.030. Fasting leptin for males: 10.1 ± 2.4 vs. 10.6 ± 2.3 ng/mL, *p* = 0.458 Designed to study sex-specific differences.	TSD effect on leptin for females: − TSD effect on leptin for males: None
Simpson, Banks, and Dinges (2010) [41]	USA	Age: 22–45 years, BMI: 17.7–32.6 kg/m^2^	145 (sleep restriction, *n* = 136; control, *n* = 9); among the SR group (*n* = 136): males = 51%, females = 49%	SR participants: 2 nights (10 h TIB/night), then 5 nights of SR (4 h TIB/night). Of these participants, 27 were randomized to receive 2 additional nights of further SR (0, 2, or 4 h TIB/night), and 37 were randomized to receive 2 nights of increased sleep time (6, 8, or 10 h TIB/night). Control subjects: 10 h TIB on all study nights	Three meals per day plus an optional evening snack; Additional snack food was available ad libitum throughout the study. Participants choose their meals’ timing (EI and EE were not assessed)	NA	In lab	Randomized parallel group study	SR vs. normal sleep: Mean leptin level for both sexes: 10.51 (±8.83) ng/mL vs. 7.88 (±6.59) ng/mL, respectively; Z = −8.43, *p* < 0.001. Mean leptin level for males: Z = −5.87, *p* = < 0.001. Mean leptin level for females: Z = −6.07, *p* = < 0.001. Designed to study sex-specific differences.	SR effect on leptin in females: + SR effect on leptin in males: + A higher effect was observed in females
Insulin										
Rao, M (2015) [44]	USA	Age: 18–45 years, BMI: 24.1 ± 4.1 kg/m^2^	14 (males, *n* = 8; females, *n* = 6)	2 nights of acclimatization, followed by 5 nights of either normal sleep (8 h TIB) or SR (4 h TIB)	An isocaloric metabolic diet consisting of fixed caloric content and proportions of carbohydrate (55%), protein (15%), and fat (30%) Fixed meal times.	4–10 weeks	In lab	Randomized crossover trial	SR vs. normal sleep: Whole-body insulin sensitivity for both sexes decreased by 25% (*p* = 0.008). No significant difference between males and females, but males had the most robust decrease in insulin sensitivity (≥30% decrease in whole-body insulin sensitivity). Not designed to study sex-specific differences.	SR effect on whole-body insulin sensitivity in females: − SR effect on whole-body insulin sensitivity in males: − The decrease was more robust in males
St-Onge, MP (2012) [42]	USA	Age: 30–45 years, BMI: 22–26 kg/m^2^	27 (males, *n* = 14; females, *n* = 13)	5 nights of SR (4 h/night) versus 5 nights of habitual sleep (9 h/night)	Controlled diet (30% fat, 55% carbohydrates, and 15% protein) with fixed meal times. Energy requirements were estimated using the Harris-Benedict equation.	3 weeks	In lab	Randomized crossover trial	SR vs normal sleep: Fasting insulin for females: lower level (regression coefficient ± SEM: −1.39 ± 0.62, *p* = 0.026). Fasting insulin for males: No significant difference. Designed to study sex-specific differences.	SR effect on fasting insulin in females: − SR effect on fasting insulin in males: None
Wang, X (2016) [43]	USA	Age: 20.6 ± 1.3 years, BMI: 24.5 ± 3.4 kg/m^2^	15 (males, *n* = 7; females, *n* = 8)	Each participant had 3-day SR (usual self-reported TIB reduced by 1–3 h) and 3-day ad libitum sleep	Ad libitum	2 weeks	Free living	Randomized crossover trial	SR vs. normal sleep: Insulin for both sexes: higher levels (*p* = 0.034) Not designed to study sex-specific differences.	SR effect on insulin in females: + SR effect on insulin in males: + No sex differences
Ghrelin										
Markwald, R (2013) [45]	USA	Age: 22.4 ± 4.8 years, BMI: 22.9 ± 2.4 kg/m^2^	16 (males, *n* = 8; females, *n* = 8)	Three baseline days of 9 h/night sleep then half participants underwent 5 days of SR (5 h/night) and the other half remained on a 9 h per night sleep. Then participants crossed over to the other condition	Weight maintenance diets during baseline period Ad libitum intake during sleep intervention	ND	In lab	Randomized cross over trial	SR vs normal sleep: Average 24 h ghrelin levels for both sexes: 660.2 ± 235.4 pg/mL vs. 794.6 ± 233.8 pg/mL, respectively. Not designed to study sex-specific differences.	SR effect on ghrelin in females: − SR effect on ghrelin in males: − No sex differences
St-Onge, MP (2012) [42]	USA	Age: 30–45 years, BMI: 22–26 kg/m^2^	27 (males, *n* = 14; females, *n* = 13)	5 nights of SR (4 h/night) versus 5 nights of habitual sleep (9 h/night)	Controlled diet (30% fat, 55% carbohydrates, and 15% protein) with fixed meal times. Energy requirements were estimated using the Harris-Benedict equation.	3 weeks	In lab	Randomized crossover trial	SR vs normal sleep: Fasting ghrelin for females: no significant difference. Fasting ghrelin for males: higher ghrelin levels (47.4 ± 24.4 pg/mL, *p* = 0.054) 24 h total ghrelin for females: no significant difference. 24 h total ghrelin for males: higher morning levels (42.5 ± 20.8 pg/mL, *p* = 0.042). Fasting active ghrelin for both sexes: no significant differences. Designed to study sex-specific differences.	SR effect on fasting ghrelin in females: None SR effect on fasting ghrelin in males: + SR effect on 24 h ghrelin (morning levels) in males: + SR effect on 24 h ghrelin (morning levels) in females: None SR effect on fasting active ghrelin in both sexes: None with no sex difference
Van Egmond, L (2023) [40]	Sweden	Age: 24.9 ± 2.9 years, BMI: 27.8 ± 6.7 (males) vs. 27.0 ± 6.1 kg/m^2^ (females)	44 (males, *n* = 24; females, *n* = 20)	Each participant had 1 night of TSD and 1 night of normal sleep	Standardized dinner followed by an overnight fast	1 week	In lab	Randomized crossover trial	TSD vs normal sleep: Fasting ghrelin for males: 703.6 ± 56.6 vs. 616.2 ± 56.1 pg/mL, *p* = 0.024 Fasting ghrelin for females: 988.8 ± 145.3 vs. 879.1 ± 111.2 pg/mL, *p* = 0.049 Designed to study sex-specific differences.	TSD effect on ghrelin in females: + TSD effect on ghrelin in males: + No sex differences
Adiponectin										
St-Onge, MP (2012) [42]	USA	Age: 30–45 years, BMI: 22–26 kg/m^2^	27 (males, *n* = 14; females, *n* = 13)	5 nights of SR (4 h night) versus 5 nights of habitual sleep (9 h/night)	Controlled diet (30% fat, 55% carbohydrates, and 15% protein) with fixed meal times. Energy requirements were estimated using the Harris-Benedict equation.	3 weeks	In lab	Randomized crossover trial	SR vs normal sleep: Fasting adiponectin for both sexes: no significant difference. 24 h adiponectin: lower levels in males (*p* = 0.0061), not in females. Designed to study sex-specific differences.	SR effect on fasting adiponectin in both sexes: None (no sex differences) SR effect on total adiponectin in males: − SR effect on total adiponectin in females: None
Van Egmond, L (2023) [40]	Sweden	Age: 24.9 ± 2.9 years, BMI: 27.8 ± 6.7 (males) vs. 27.0 ± 6.1 kg/m^2^ (females)	44 (males, *n* = 24; females, *n* = 20)	Each participant had 1 night of TSD and 1 night of normal sleep	Standardized dinner followed by an overnight fast	1 week	In lab	Randomized crossover trial	TSD vs normal sleep: Fasting adiponectin for males: 5.9 ± 0.5 vs. 5.6 ± 0.6 μg/mL, *p* = 0.056 Fasting adiponectin for females: 9.4 ± 1.0 vs. 8.4 ± 0.9 μg/mL, *p* = 0.025 Designed to study sex-specific differences.	TSD effect on adiponectin in females: + TSD effect on adiponectin in males: None
GLP−1										
St-Onge, MP (2012) [42]	USA	Age: 30–45 years, BMI: 22–26 kg/m^2^	27 (males, *n* = 14; females, *n* = 13)	5 nights of SR (4 h/night) versus 5 nights of habitual sleep (9 h/night)	Controlled diet (30% fat, 55% carbohydrates, and 15% protein) with fixed meal times. Energy requirements were estimated using the Harris-Benedict equation.	3 weeks	In lab	Randomized crossover trial	SR vs normal sleep: Fasting total GLP-1 for both sexes: no significant differences. 24 h GLP-1 for females: lower afternoon levels (*p* = 0.016). Designed to study sex-specific differences.	SR effect on fasting total GLP-1 in both sexes: None (no sex differences) SR effect on 24 h GLP-1 in females: − (afternoon levels) SR effect on 24-h GLP-1 in males: None
PYY										
Markwald, R (2013) [45]	USA	Age: 22.4 ± 4.8 years, BMI: 22.9 ± 2.4 kg/m^2^	16 (males, *n* = 8; females, *n* = 8)	Three baseline days of 9 h/night sleep, then half the participants underwent 5 days of SR (5 h/night), and the other half remained on a 9 h per night sleep. Then participants crossed over to the other condition	Weight maintenance diets during baseline period Ad libitum intake during sleep intervention	ND	In lab	Randomized crossover trial	SR vs normal sleep: Average 24 h PYY for both sexes: 136.1 ± 44.8 pg/mL vs. 100.5 ± 35.1 pg/mL, respectively. Not designed to study sex-specific differences.	SR effect on PYY in females: + SR effect on PYY in males: + No sex differences
St-Onge, MP (2012) [42]	USA	Age: 30–45 yrs, BMI: 22–26 kg/m^2^	27 (males, *n* = 14; females, *n* = 13)	5 nights of SR (4 h/night) versus 5 nights of habitual sleep (9 h/night)	Controlled diet (30% fat, 55% carbohydrates, and 15% protein) with fixed meal times. Energy requirements were estimated using the Harris-Benedict equation.	3 weeks	In lab	Randomized crossover trial	SR vs normal sleep: Fasting PYY for both sexes: no significant differences. Designed to study sex-specific differences.	SR effect on fasting PYY in females: None SR effect on fasting PYY in males: None No sex differences
Glucagon										
Wang, X (2016) [43]	USA	Age: 20.6 ± 1.3 years, BMI: 24.5 ± 3.4 kg/m^2^	15 (males, *n* = 7; females, *n* = 8)	Each participant had 3-day SR (usual self-reported TIB reduced by 1–3 h) and 3-day ad libitum sleep	Ad libitum	2 weeks	Free living	Randomized crossover trial	SR vs. normal sleep: Fasting glucagon for both sexes: higher levels (*p* = 0.003) Not designed to study sex-specific differences.	SR effect on glucagon in females: + SR effect on glucagon in males: + No sex differences
T3 and T4										
Kessler, L (2010) [46]	USA	Age: 39 ± 5 years, BMI: 26.5 ± 1.5 kg/m^2^	11 (males, *n* = 6; females, *n* = 5) (6 for rT3)	Each subject completed two 14-day intervention periods with sedentary activity, ad libitum food intake, and scheduled time-in-bed of 5.5 (SR) or 8.5 h/night in random order	Ad libitum intake. EI > EE during both sleep conditions	At least 3 months	In lab	Randomized crossover trial	SR vs. normal sleep: fT4 for males: 0.98 ± 0.06 vs. 1.06 ± 0.05 mcg/dL; *p* = 0.14. fT4 for females: 1.10 ± 0.03 vs. 1.19 ± 0.05 mcg/dL; *p* < 0.001 rT3 for both sexes: no significant difference Not designed to study sex-specific differences.	SR effect on fT4 in females: − SR effect on fT4 in males: None SR effect on rT3 in both sexes: None No sex differences
Petrov, M (2024) [47]	USA	Age: 36.2 years ± 12.8 yrs, BMI: 26.7 ± 3.1 kg/m^2^	30 (males, *n* = 10; females, *n* = 20)	Each subject completed 6 weeks of normal sleep and SR (sleep duration reduced by 1.5 h from the usual sleep duration).	Ad libitum	2–6 weeks	Free living	Randomized crossover trial	SR vs. normal sleep: FT4 for females: non-significant increases (β = 0.08 ± 0.06 ng/dL, *p* = 0.177, Cohen’s *f^2^* = 0.05) FT4 for males: non-significant decreases (β = −0.05 ± 0.03 ng/dL, *p* = 0.127, Cohen’s *f^2^* = 0.08) Designed to study sex-specific differences.	SR effect on FT4 in females: + SR effect on FT4 in males: −

SEM: Standard Error of the Mean; SR: sleep restriction; TSD: total sleep deprivation; TIB: time in bed; EI: energy intake; EE: energy expenditure; ND: Not Determined; NA: Not Applicable.

## Data Availability

No new data were created or analyzed in this study. Data sharing is not applicable to this article.

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
