# Peer review of "The Role of Sex in the Impact of Sleep Restriction on Appetite- and Weight-Regulating Hormones in Healthy Adults: A Systematic Review of Human Studies"

_2624-5175, 2025, doi:10.3390/clockssleep7030039_

Round 1
Reviewer 1 Report
Comments and Suggestions for Authors
Dear Authors,
Thank you for the opportunity to read your systematic review. The topic is interesting and adds to the much-required literature that investigates sex differences of appetite markers in response to sleep restriction. Whilst the literature shows inconsistencies in the response to sleep restriction, you have highlighted several reasons which should be investigated further.
I would like to congratulate the authors on a well written systematic review. I do have some minor comments to help improve the manuscript.
Abstract:
I think it would be useful to outline how the data were analysed; this would provide some context of any differences.
When including the number of studies, there were inconsistencies in how this was reported. For example, leptin (four studies), insulin (3 studies), glucagon like peptide-1 32 (one study), ghrelin (3 studies), adiponectin (2 studies). Ensure you are consistent with either numerical or fully written numbers. I believe 1-9 should be spelled out.
Introduction
The introduction is well written, but I feel that the reader would benefit from a brief introduction to the appetite markers of interest, and how they act to influence appetite. Particularly lesser studied appetite markers where mechanisms have not been outlined in the discussion section.
Methods
In section 2.1 of the methods, you state that PsycINFO was used to search for literature, but these data are not included in the PRISMA flow chart.
Results
Table 1. Perhaps presenting this table in landscape would improve the readability.
Table 1. Rao (2015). Could you clarify what is meant by ‘the decrease was more robust in males’? Including numerical values would be useful.
Discussion
Line 240: Ensure ‘Discussion’ is formatted as a heading.
Reviewer 2 Report
Comments and Suggestions for Authors
The review investigates sex-specific alterations in appetite- and weight-regulating
hormones in response to sleep restriction. The manuscript is well structured and
addresses a relevant topic concerning sleep health and its impact on hormones
involved in energy metabolism, highlighting the different responses in males and
females. The authors found inconclusive results for leptin, insulin, GLP-1, ghrelin,
adiponectin and T4, and no influence of sex on glucagon or PYY after sleep restriction. Some aspects should be addressed to improve the clarity of the manuscript.
Major Concerns
There is inconsistency in the use of the abbreviation 'SR' throughout the manuscript. Initially, 'SR' refers to 'Sleep Restriction', but later it is used for 'Stress Response', and the abbreviation for 'Sleep Restriction' is no longer used. Please revise the manuscript to ensure consistent and unambiguous use of the abbreviation 'SR'.
1. Figure 1
There seems to be information cut off in the chart, particularly in the box
referring to ‘Records excluded’.
Minor Concerns
1. Table 1
The abbreviation ‘TIB’ in the table is not defined in the legend. Please provide
the full term for clarity.
2. Discussion
a. The Discussion section title is not properly formatted; it is not bold and
appears on the same line as the first paragraph. Please correct this
formatting issue.
b. The abbreviation for ‘hypothalamic-pituitary-adrenal (HPA) axis’ is
explained twice in the same paragraph. Please keep only the first
instance and remove the repetition (line 300).
3. List of abbreviations
The abbreviation 'SR' (Sleep Restriction) is missing from the list of
abbreviations. Please include it.
Reviewer 3 Report
Comments and Suggestions for Authors
The study by AlFikany et al. evaluates the results of several independent studies to uncover a connection between the hormonal response to sleep restriction in relation to males and females. The research question is current and relevant. However, its relevance to everyday life should be made more clear (see below).
When evaluating the study results, considerably more emphasis should be placed on the general characteristics of the subjects, such as age and BMI. These likely explain the discrepancies between the study results (see below).
Introduction/Discussion:
Please mention the relevance of sleep deprivation for everyday situations, e.g. in which people who start school or work early are also exposed to a certain degree of constant sleep deprivation, at least on weekdays, especially people with a later chronotype, such as adolescents.
Discussion:
- I agree, that the data indicates that sex alone is insufficient to account for the discrepancies in the hormonal responses to SR and additional variables should be considered. At the beginning of the discussion, possible explanations for the discrepancies between the study results should be discussed.
1a. BMI is a very important variable in this context and shows a high degree of variation among the cited studies. According to the WHO classification, a BMI of >25 kg/m2 is considered pre-obesity, and a BMI of >30 kg/m2 is considered Class I obesity. Perhaps only studies that do not include subjects with a BMI above normal weight reflect a physiological hormonal response to SR. Studies including subjects with pre-obesity and obesity likely reflect a response in a system in which metabolism is already somewhat derailed. I highly recommend re-evaluating the data in this light.
1b. The age of the subjects is also an important variable, as resilience to sleep deprivation is age-dependent and also shows considerable differences between studies.
- The connection between SR and stress is highly relevant. Are there any studies that support the assumption that the HPA axis might play a crucial role as a mediator in the sex-dimorphic relationship between sleep deprivation and appetite regulation? The study you cite was conducted only in men.
Minor: Please keep the sex order consistent when naming the effects (e.g. always first female, then male).
Lines 255 Stress response is abbreviated as SR. However, this is short for sleep restriction. Please do not use these terms synonymously.
Round 2
Reviewer 3 Report
Comments and Suggestions for Authors
none
Author Response
Thank you for reviewing our revised manuscript and for your valuable feedback throughout the review process.
There are no additional comments to address at this stage.